# A Lightweight Building Extraction Approach for Contour Recovery in Complex Urban Environments

**Jiaxin He** [1], **Yong Cheng** [2], **Wei Wang** [1], **Zhoupeng Ren** [3,*], **Ce Zhang** [4] and **Wenjie Zhang** [5]

1    School of Automation, Nanjing University of Information Science & Technology, Nanjing 210044, China
2    School of Software, Nanjing University of Information Science & Technology, Nanjing 210044, China
3    State Key Laboratory of Resources and Environmental Information System, Institute of Geographic Sciences and Natural Resources Research, Chinese Academy of Sciences, Beijing 100101, China
4    School of Geographical Sciences, University of Bristol, Bristol BS8 1SS, UK; ce.zhang@bristol.ac.uk
5    School of Geographical Sciences, Nanjing University of Information Science & Technology, Nanjing 210044, China
*    Correspondence: renzp@lreis.ac.cn

**Abstract:** High-spatial-resolution urban buildings play a crucial role in urban planning, emergency response, and disaster management. However, challenges such as missing building contours due to occlusion problems (occlusion between buildings of different heights and buildings obscured by trees), uneven contour extraction due to mixing of building edges with other feature elements (roads, vehicles, and trees), and slow training speed in high-resolution image data hinder efficient and accurate building extraction. To address these issues, we propose a semantic segmentation model composed of a lightweight backbone, coordinate attention module, and pooling fusion module, which achieves lightweight building extraction and adaptive recovery of spatial contours. Comparative experiments were conducted on datasets featuring typical urban building instances in China and the Mapchallenge dataset, comparing our method with several classical and mainstream semantic segmentation algorithms. The results demonstrate the effectiveness of our approach, achieving excellent mean intersection over union (mIoU) and frames per second (FPS) scores on both datasets (China dataset: 85.11% and 110.67 FPS; Mapchallenge dataset: 90.27% and 117.68 FPS). Quantitative evaluations indicate that our model not only significantly improves computational speed but also ensures high accuracy in the extraction of urban buildings from high-resolution imagery. Specifically, on a typical urban building dataset from China, our model shows an accuracy improvement of 0.64% and a speed increase of 70.03 FPS compared to the baseline model. On the Mapchallenge dataset, our model achieves an accuracy improvement of 0.54% and a speed increase of 42.39 FPS compared to the baseline model. Our research indicates that lightweight networks show significant potential in urban building extraction tasks. In the future, the segmentation accuracy and prediction speed can be further balanced on the basis of adjusting the deep learning model or introducing remote sensing indices, which can be applied to research scenarios such as greenfield extraction or multi-class target extraction.

**Keywords:** remote sensing images; lightweight; context information; adaptive recovery; building extraction

## 1. Introduction

Efficient extraction of buildings from remote sensing images can provide geospatial data of buildings with wide coverage, clear spatial information, and fast update speed for urban planning, disaster management, and other scenarios [1–7]. With the continuous progress of remote sensing technology, researchers use building feature information in various multi-temporal high-resolution remote sensing images to discriminate and manually extract urban buildings by visual interpretation and manual labelling [8,9]. However, due to different lighting conditions, some non-buildings (containers, cars, and roads) may have

similar spectral and spatial features to buildings; visual interpretation may then misjudge buildings, which leads to mislabeling of non-buildings by manual annotation, resulting in poor building extraction [10–12]. Therefore, researchers are trying to use better methods to improve the building extraction accuracy.

Presently, researchers primarily extract building information from remote sensing images using both traditional and deep learning methods. Traditional methods for urban building extraction from remote sensing images include clustering algorithms, support vector machines, and random forests, among others [13–15]. Gavankar et al. devised an object-based approach that leverages high-resolution multispectral satellite images, combining K-means clustering and shape parameters to extract building outlines [16]. While this approach combines pixel-level information with object-level features, thus improving the accuracy of building extraction, clustering algorithms may struggle to differentiate between buildings and other features when they share similar spectral characteristics in complex background environments. Arham et al. explored object-based image analysis methods, combining support vector machines (SVMs) with rule-based image classification for building extraction tasks [17]. SVMs can provide relatively high precision in building extraction, especially with appropriate feature engineering and parameter tuning, even in complex urban settings. However, an SVM's performance can be limited by factors such as noise and occlusions (e.g., trees, clouds, and tall buildings), especially when extracting fine-grained building outlines. Chen et al. employed a method based on random forests and superpixel segmentation to automatically extract buildings from remote sensing data [18]. While random forests exhibit robustness to noisy data and missing information, they struggle to provide detailed internal decision rules in complex urban scenes, resulting in reduced interpretability and potentially affecting the accuracy of urban building extraction. The performance of these three traditional methods heavily relies on the chosen feature sets, and inappropriate or incomplete feature selection can lead to suboptimal building extraction results. Therefore, a more precise research approach is needed to enhance the accuracy of urban building extraction.

With the rapid development in the field of computer vision, researchers have started applying deep learning methods to urban building extraction tasks [19]. Deep learning, based on neural network structures, autonomously learns the relevant features of buildings in large-scale high-resolution remote sensing images (such as spectral features, scale features, and texture features), enabling efficient and accurate building extraction [20]. Currently, deep learning-based building extraction methods have been widely applied in areas such as object detection and semantic segmentation [21,22]. However, while object detection methods can successfully detect buildings, they are unable to extract more detailed urban building contours. Therefore, researchers have chosen to employ deep learning-based semantic segmentation methods for urban building extraction. In 2016, through the encoder–decoder structure, Zhong et al. applied a fully convolutional network (FCN) to extract buildings from high-resolution remote sensing images to achieve pixel-level segmentation [23]. Subsequently, derived semantic segmentation methods such as PSPNet [24], U-Net [25], HRNet [26], and the Deeplab series [27–30] have been utilized to further enhance building extraction efficiency [31,32]. Aiming to solve the problems of insufficient high-precision building datasets and the inability of semantic segmentation to further classify buildings, Ji et al. established a large-scale, high-precision building dataset (WHUbuildingdataset) covering multiple sample forms (raster and vector) and multiple data sources (aviation and satellites), and achieved the identification and extraction of buildings through an instance segmentation method based on Mask R-CNN [33]. Since traditional building extraction methods make it difficult to accurately segment buildings, roads, and trees in complex scenes, Xu et al. proposed a multi-layer feature fusion dilated convolution ResNet model, effectively overcoming interference from non-building objects like trees and roads [34]. Considering the problems of low segmentation accuracy and blurred edge contour in traditional building extraction methods, Zhang et al. proposed the combination of a U-net neural network and a fully connected CRFs network and opti-

mized the segmentation results according to image features, which significantly improved the segmentation accuracy and building contour integrity [35]. Yang et al. adopted the Deeplabv3plus algorithm to enhance the expression ability of building detail information, and compared the classification performance of Deeplabv3, Deeplabv3plus and BiseNet by using the building sample library, which solved the problem that machine learning has poor robustness in building extraction tasks and finds it difficult to fully mine the deep features of buildings [36,37]. Although these methods improve accuracy and efficiency to some extent, building features are not easily extracted due to problems such as buildings being occluded by other ground features (high-rise buildings and trees) and building edges being mixed with other non-building elements (roads, vehicles, and trees), which in turn result in partially missing building outlines. Therefore, a deep learning method is needed to avoid the occlusion and mixing problems arising between non-building feature elements and buildings and to maintain the integrity of building contours for the task of urban building extraction.

In conclusion, we propose an improved deep learning network to address the issues of incomplete contours in the building extraction process. To tackle the problem of missing building contours, we introduced the coordinate attention module into the improved network. By learning the coordinate information of different positions, we enhanced the attention given to the accurate positioning of building space, and then improved the accuracy of building edge contour extraction. In order to further optimize the building contour, we designed a pooling fusion module to improve the clarity of the building contours and enhance the perception of the overall structure of the building, so as to achieve a comprehensive optimization of the building contour. It is worth mentioning that the current method is low in computational efficiency and cannot support large-scale urban building extraction applications. To achieve fast extraction of urban buildings, we employ a lightweight backbone network to enhance the model's inference efficiency by reducing the number of model parameters. The main contributions of this research are as follows:

(1) We propose an advanced deep learning-based method for extracting urban buildings from high-resolution remote sensing images. In this study, we replaced the backbone network by adopting a lightweight model to address the issue of low computational efficiency in existing models. Additionally, we introduced an attention mechanism to enhance the focus on the spatial coordinate information of building contours at different locations in the image, aiming to alleviate the problem of missing architectural outlines.

(2) Our improved network incorporates a fusion module that combines strip pooling with Atrous spatial pyramid pooling to introduce lateral context information to further recover building contour profiles by enhancing the network's representation of building edge features. We validate the significant role of strip pooling in enhancing the feature extraction of urban buildings.

## 2. Methods

### 2.1. The Structure of the Network

This paper proposes an improved deep learning network for urban building extraction from high-resolution remote sensing images as shown in Figure 1. Firstly, the backbone network Mobilenetv2 is introduced to reduce the scale of model parameters and improve the inference speed of the model [38]. Additionally, a coordinate attention (CA) module [39] is introduced into the improved model to enhance the attention given to the spatial coordinate information of the building contour. This helps to make the network more flexible to adapt to the contour features of buildings at different locations, thereby optimizing the contour extraction results. Second, the pooling fusion module combines strip pooling [40] (SP) and dilated spatial pyramid pooling to obtain more comprehensive background information in both horizontal and vertical directions to optimize the fine features of the building contours and the overall structural perception. Finally, the feature map formed after pooling fusion is downscaled by $1 \times 1$ convolution and a new feature map is formed by 4-fold

upsampling, while the backbone feature extraction network Mobilenetv2 is downscaled by $1 \times 1$ convolution to form another feature map; then, the two newly formed feature maps mentioned above are fused, and a building contour map with the same dimensions of the original image is generated by using $3 \times 3$ convolution with 4-fold upsampling.

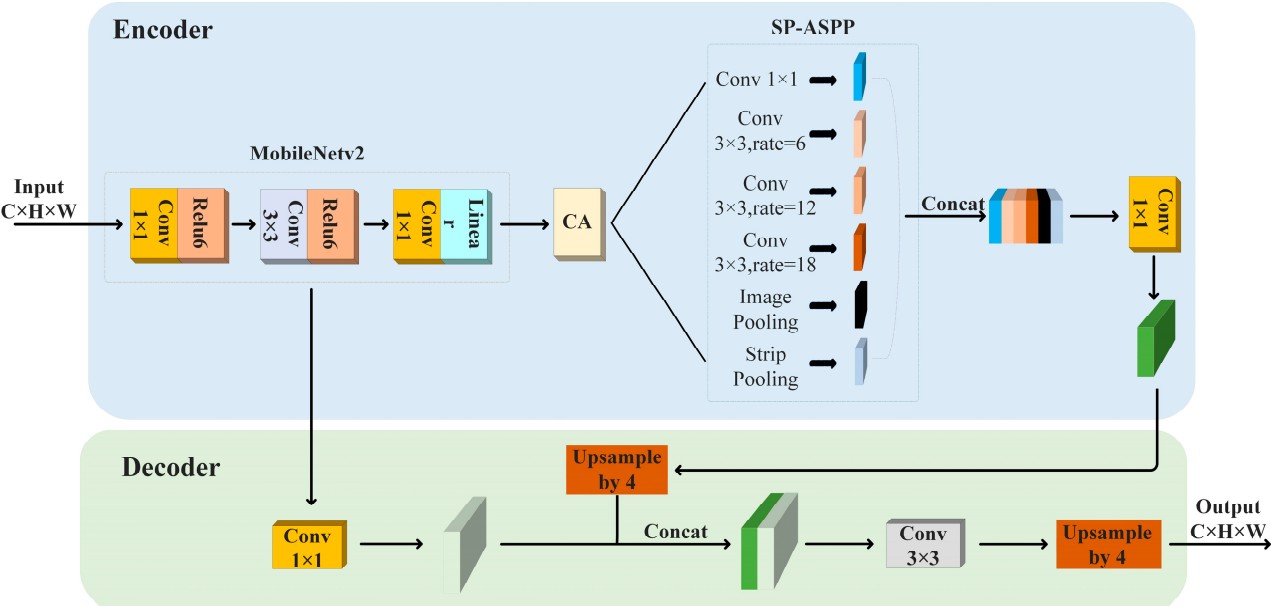

**Figure 1.** Improved network structure.

## 2.2. Coordinate Attention Mechanism

In the task of extracting urban buildings from high-resolution remote sensing images, there are occlusion problems for buildings in certain areas (occlusion between buildings of different heights and occlusion of low-height buildings by trees) due to interferences in shadows, lighting conditions, etc., which makes it difficult for the model to accurately extract the outlines of buildings in these areas. When dealing with such problems, the existing network is unable to optimise the effect of building contours in these areas, resulting in uneven extraction of building contours. Therefore, this study introduces a coordinate attention (CA) module into the DeepLabv3plus to address this problem. The CA module helps to improve the model, making it more adaptable to the characteristics of the building contours at different locations, thus optimising the building contours in these areas.

The coordinate attention (CA) module is a technique used to enhance the relationship between input feature channels in a neural network. It leverages the coordinate information of each location in the input image to calculate attention weights and obtain a comprehensive feature representation of the entire image. Firstly, the input image is partitioned into multiple grids, each encompassing positional information. Then, the positional information of each grid is used as input to compute its corresponding weights using a simple multi-layer perceptron or a more complex neural network. Finally, the computed weights are used to find weighted averages of the feature representations of each grid, resulting in the feature information of all targets in the entire image. By utilizing position encoding, CA focuses on different regions at various positions in the image, capturing the spatial information within the image. The CA module is depicted in Figure 2.

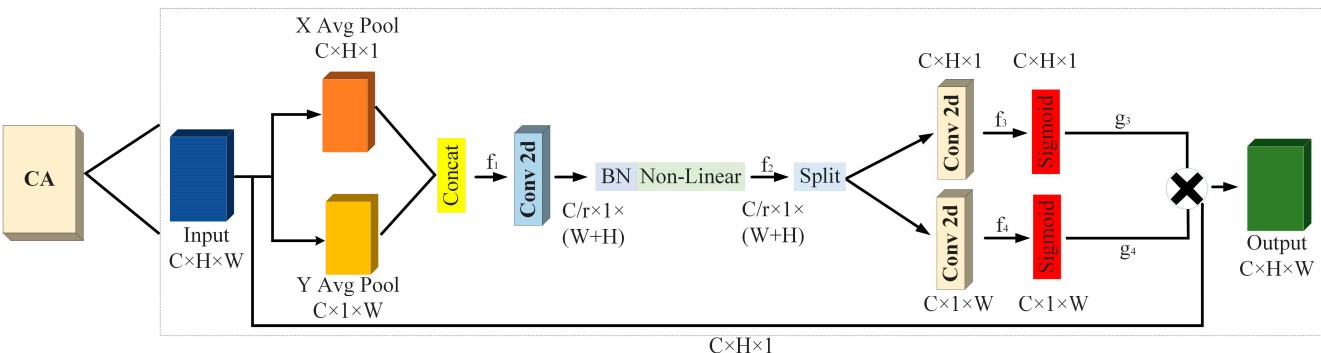

**Figure 2.** Coordinate attention module.

Based on Figure 2, the process can be described as follows. Initially, the input feature map "Input" of size C × H × W undergoes pooling operations along the X and Y directions, resulting in feature maps of size C × H × 1 and C × 1 × W, respectively. Subsequently, the feature maps of size C × 1 × W and C × H × 1 undergo transformations and are fused to generate the feature map "$f_1$." Next, "$f_1$" is subjected to dimension reduction and activation using a 1 × 1 convolutional kernel, yielding the feature map "$f_2$." Finally, "$f_2$" is spatially segmented (split) into two feature maps, "$f_3$" and "$f_4$." These feature maps are then upsampled using 1 × 1 convolutions and combined with the Sigmoid activation function to obtain attention vectors "$g_3$" and "$g_4$,", respectively.

### 2.3. SP-ASPP Module

To address the problem of degradation of building contour extraction accuracy due to the mixing of building edges with other feature elements (roads, vehicles, and trees), the SP-ASPP module is introduced in this paper. The SP-ASPP module integrates a strip pooling (SP) branch [40] after the average pooling layer within the ASPP module. By introducing lateral context information, it helps to better capture the edge features of the building and thus enhances the recovery of the building contours. The SP-ASPP module is illustrated in Figure 3.

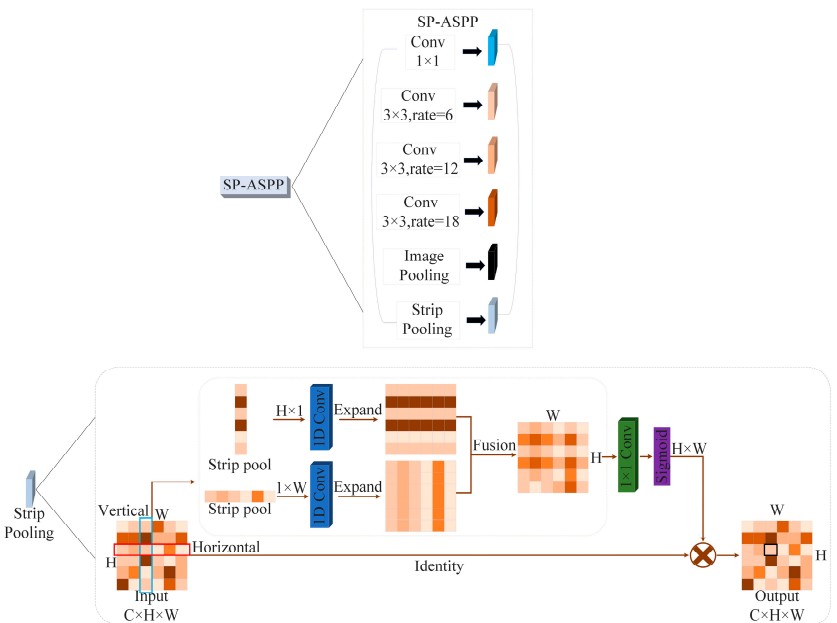

**Figure 3.** SP-ASPP module.

Based on Figure 3, the process can be described as follows. Firstly, the input feature map is subjected to dimension reduction using a 1 × 1 convolution, resulting in feature

vectors that enhance the model's capability to fuse diverse channel features. Subsequently, the feature vectors are processed through $3 \times 3$ dilated convolutions with dilation rates of 6, 12, and 18, respectively, to extract features under different sensory fields. By using different rates of expansion to form different sensory fields, the convolutions effectively expand the receptive fields without introducing additional parameters, thereby capturing a wider range of context information in the output feature maps. Next, for the multi-layer feature maps, pooling operations are employed to obtain fixed-sized feature vectors that encompass multi-level context information, facilitating a more comprehensive capture of global semantic information in the image. Finally, a strip pooling branch is introduced, which partitions the input feature map into multiple strip regions and performs pooling, leading to an increased channel dimension of the feature map and an enhanced representational capacity of the model.

## 3. Experiment

### 3.1. Experimental Environment

The experimental setup involves using the Windows 10 operating system, an NVIDIA GeForce RTX 3070Ti GPU, and 8 GB of memory. The deep learning framework utilized is PyTorch 1.7.1 with CUDA 11.6. For this experiment, pretrained weights from the VOC dataset are employed, and the training process is divided into two phases: the freezing phase and the thawing phase. The model's optimizer is stochastic gradient descent (SGD), and the learning rate is adjusted using the cosine annealing method, which helps optimize the training process over iterations. To enhance the available data, the Mixup and Mosaic methods are employed, which provide data augmentation techniques to improve the model's generalization and robustness by combining and manipulating training samples. These techniques collectively contribute to the performance improvement and accuracy enhancement of the model in object detection tasks.

### 3.2. Dataset

(1) The primary dataset employed in this study is the publicly available Remote Sensing Image Dataset of Typical Urban Buildings in China [41], which was curated by China University of Geosciences. The dataset contains 7260 image area samples and a total of 63,886 buildings; four representative urban centers, Beijing, Shanghai, Shenzhen, and Wuhan, were selected as the data collection target areas. The original data were sourced from 19th-level satellite imagery provided by Google, with a ground resolution of 0.29 m. To ensure the dataset's versatility, the selection of data regions considered diverse factors such as the inclusion of orthorectified and non-orthorectified image regions, areas with both sparse and dense distributions of buildings, and the incorporation of various building contour shapes. The dataset was partitioned into training, validation, and testing sets, following an 8:1:1 ratio, and the image dimensions were standardized to $500 \times 500$ pixels. The dataset was exported in the Pascal VOC data format. Sample images from the dataset are depicted in Figure 4.

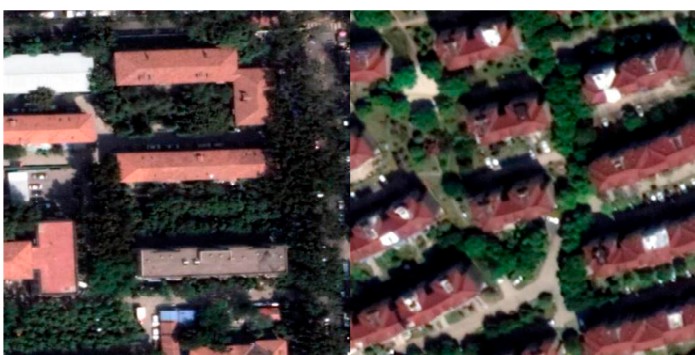

**Figure 4.** Examples of typical urban buildings in China.

(2) The second dataset employed in this study is the Map Challenge Building Dataset [42]. The images within this dataset have dimensions of 300 × 300 pixels. It encompasses a total of 280,741 training samples, 60,697 testing samples, and 60,317 validation samples. For the purpose of this study, 6028 images were selected from this dataset. These images were divided into training, validation, and testing sets following an 8:1:1 ratio, and the data were exported in the Pascal VOC data format. Sample images from the dataset are depicted in Figure 5.

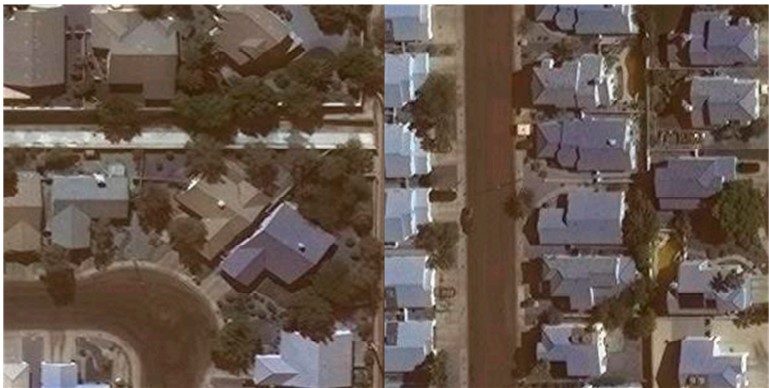

**Figure 5.** Mapchallenge building dataset.

*3.3. Evaluation Index*

In order to adequately assess the performance of the enhanced model, qualitative evaluations were carried out using mean Intersection over Union (mIoU), pixel accuracy, mean pixel accuracy, and Intersection over Union (IoU), as defined in Equations (1)–(4), respectively. Mean Intersection over Union (mIoU) denotes the average of the ratio of the intersection and concatenation of the model's predictions for the two categories of building and background to the actual labeling, which is used to evaluate segmentation performance in a comprehensive way. pixel Accuracy denotes the ratio of the number of pixels of a building correctly predicted by the model to the total number of pixels. Mean pixel accuracy in denotes the average ratio between the model's correct prediction of pixels in both building and background categories. Intersection over Union (IoU) denotes the ratio of intersection over union between the model's predicted and actual building regions for evaluating segmentation accuracy. The operational performance of the algorithm was measured in terms of frames per second (FPS), which represents the number of frames processed per second.

Based on the above equations, TP denotes the number of true positive samples accurately predicted for a specific class, FP represents the number of false positive samples incorrectly classified as positive for a specific class, FN indicates the number of false negative samples incorrectly classified as negative for a specific class, and TN signifies the number of true negative samples accurately predicted for a specific class. The variable 'n' represents the total number of classes.

$$mIoU = \frac{1}{n+1}\sum_{i=0}^{n}\frac{TP}{TP+FP+FN} \tag{1}$$

$$PA = \frac{TP+TN}{TP+TN+FP+FN} \tag{2}$$

$$mPA = \frac{1}{n+1}\sum_{i=0}^{n}\frac{TP+TN}{TP+TN+FP+FN} \tag{3}$$

$$IoU = \frac{TP}{TP+FP+FN} \tag{4}$$

### 3.4. Ablation Experiment

In order to verify the effectiveness of the improved strategies, a series of ablative experiments were conducted on the Remote Sensing Image Dataset of Typical Urban Buildings in China, using Deeplabv3plus as the baseline model. The experimental setup and parameters were kept consistent. Table 1 presents a comparative analysis of the architectural extraction performance using different modules, namely Mobilenet, CA (coordinate attention), and SP-ASPP (strip pooling and Atrous spatial pyramid pooling) modules. In this context, Mobilenet serves as the backbone network, CA represents the coordinate attention module, and SP-ASPP combines strip pooling with Atrous spatial pyramid pooling for pooling fusion.

**Table 1.** Ablation experiment of the improved strategies.

| Method | IoU (Building)/(%) | mIOU/(%) | PA/(%) | mPA/(%) |
| --- | --- | --- | --- | --- |
| Deeplabv3plus | 75.61 | 84.47 | 88.11 | 91.63 |
| Deeplabv3plus + Mobilenetv2 | 74.97 | 84.13 | 86.07 | 91.25 |
| Deeplabv3plus + CA | 76.02 | 84.74 | 88.39 | 92.23 |
| Deeplabv3plus + SP-ASPP | 76.27 | 84.99 | 85.21 | 91.23 |
| All | 76.44 | 85.11 | 86.35 | 91.61 |

From Figure 6 and Tables 1 and 2, it can be observed that the introduction of the Mobilenetv2 module results in a significant reduction in the number of model parameters and floating-point operations compared to the baseline network, with a 0.34% reduction in the mean accuracy (mIoU) and a 0.38% reduction in mPA. However, the network's prediction speed increased by 96.89 FPS, indicating that the Mobilenetv2 module, while potentially sacrificing architectural extraction accuracy, significantly enhances the computational efficiency of the model. The CA attention module effectively enhances the ability to adapt to the building contour features at each location, and improves the problem of missing building contours due to trees and high-rise buildings occluding low-rise buildings. The mIoU is improved by 0.27%, with a slight increase in prediction speed. The SP-ASPP module achieves an accurate distinction between building contours and non-building contours by focusing on the extraction of building edge features, resulting in a 0.52% increase in mIoU, albeit with a slight decrease in prediction speed. In summary, with respect to extraction accuracy, apart from the reduction in the Mobilenetv2 backbone network, the other two improvement modules show enhancement. Regarding prediction speed, the improved backbone network demonstrates significant improvement, while other enhancement strategies have minimal impact on prediction speed. Moreover, the fusion of the three improvement strategies yields a 0.64% mIoU gain for the model, confirming the effectiveness of the enhancement strategies.

**Table 2.** Number of parameters used to improve the strategy.

| Method | Gflops (GB) | Params (MB) | FPS/(f/s) |
| --- | --- | --- | --- |
| Deeplabv3plus | 166.84 | 54.71 | 40.64 |
| Deeplabv3plus + Mobilenetv2 | 6.26 | 2.06 | 137.53 |
| Deeplabv3plus + CA | 166.88 | 55.10 | 41.14 |
| Deeplabv3plus + SP-ASPP | 177.17 | 61.27 | 38.15 |
| All | 6.51 | 2.23 | 110.67 |

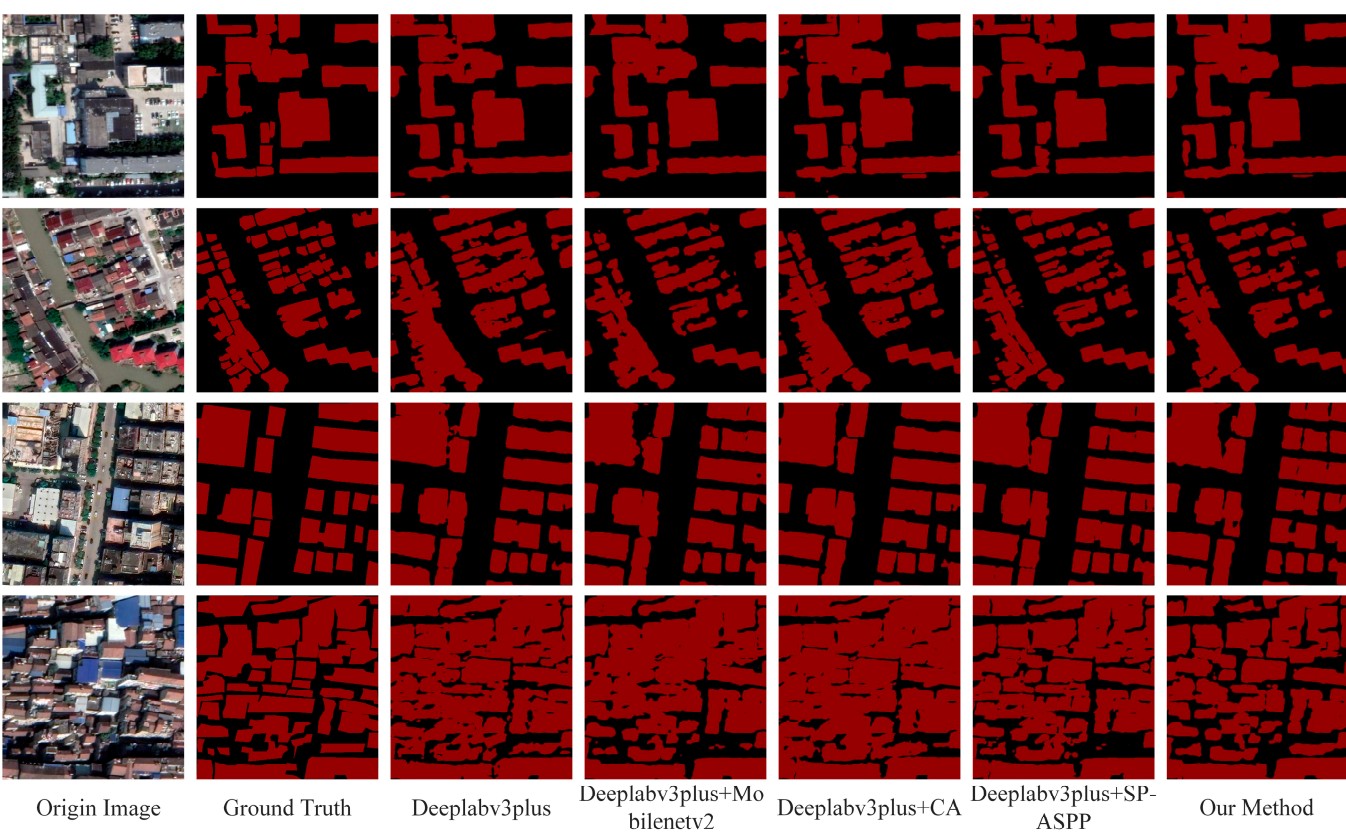

| Origin Image | Ground Truth | Deeplabv3plus | Deeplabv3plus+Mobilenetv2 | Deeplabv3plus+CA | Deeplabv3plus+SP-ASPP | Our Method |

**Figure 6.** Model visualization results on a dataset of buildings in typical Chinese cities.

### 3.5. Comparison with Other Algorithms

To assess the effectiveness and generalization of the improved model for urban building extraction, experiments were conducted on both the Mapchallenge dataset and a dataset containing typical urban building instances in China. The improved model was compared against other state-of-the-art algorithms, and their extraction accuracies were compared. For the comparative experiments on the Mapchallenge dataset, the reference models selected were PSPNet, U-Net, Deeplabv3plus, and HRNetv2. The comparative experiments demonstrated that the improved model achieved a mean intersection over union (mIoU) of 90.24%, surpassing PSPNet (Mobilenet), PSPNet, U-Net (vgg), U-Net (resnet50), HRNetv2_w32, Deeplabv3plus, HRNetv2_w48, and HRNetv2_w18 by 6.94%, 1.71%, 1.2 3%, 0.96%, 0.54%, 0.54%, 0.34%, and 0.20%, respectively, as depicted in Table 3. The comparative experiment results on the dataset of typical urban building instances in China indicated that the improved model achieved an mIoU of 85.11%, outperforming PSP-Net (Mobilenet), PSPNet, U-Net (vgg), U-Net (resnet50), HRNetv2_w32, Deeplabv3plus, HRNetv2_w48, and HRNetv2_w18 by 7.70%, 2.10%, 0.94%, 0.60%, 1.23%, 0.64%, 0.25%, and 0.31%, respectively, as depicted in Table 5. These findings demonstrate that the improved method exhibits higher extraction accuracy and faster prediction speed.

**Table 3.** Comparison experiment results (Mapchallenge building dataset).

| Models | IoU (Building)/(%) | mIoU/(%) | PA/(%) | mPA/(%) |
|---|---|---|---|---|
| PSPNet(Mob) | 74.43 | 83.33 | 84.86 | 90.49 |
| PSPNet | 82.37 | 88.56 | 89.12 | 93.56 |
| U-Net(vgg) | 83.06 | 89.04 | 89.62 | 93.90 |
| U-Net(resnet50) | 83.48 | 89.31 | 90.61 | 94.12 |
| HRNetv2_w32 | 84.12 | 89.73 | 90.58 | 94.30 |
| Deeplabv3plus | 84.13 | 89.73 | 91.18 | 94.43 |
| HRNetv2_w48 | 84.40 | 89.93 | 90.86 | 94.37 |
| HRNetv2_w18 | 84.63 | 90.07 | 91.03 | 94.47 |
| Ours | 84.86 | 90.27 | 92.22 | 94.79 |

On the MapChallenge dataset, according to Tables 3 and 4, the proposed improved model achieves the highest accuracy as well as faster speed in urban building extraction compared to other algorithms. Specifically, the mIoU reaches 90.27%, the PA is 92.22%, the mPA is 94.79%, and the FPS rate is 117.68. As depicted in Figure 7, second row, due to the fact that different terrain elements and building roofs may have low feature variability (spectra, colours), PSPNet (Mobilenet), PSPNet, U-Net (resnet50), Deeplabv3plus, and HRNetv2_w18 tend to extract redundant terrain element information, leading to ineffective segmentation of building outlines and inferior extraction results. In contrast, the proposed improvement method clearly exhibits smooth building edges without misidentifying non-building elements, closely resembling real-world surface conditions. In the first row of Figure 7, for dispersed building layouts, each network can effectively segment buildings. However, due to the occlusion between buildings of varying heights and interference from environmental elements such as trees, roads, and vehicles, the accuracy of building extraction is hindered. U-Net (resnet50), HRNet_w32, and PSPNet (Mobilenet) all exhibit instances of erroneously extracting non-building terrain elements, failing to effectively extract building parts. Furthermore, when extracting large-scale buildings, the proposed method demonstrates more pronounced refinement of building edge contours, achieving better building segmentation accuracy.

**Table 4.** Comparison of the individual model parameters in the Mapchallenge building dataset.

| Models | Gfloaps/(GB) | Params/(MB) | FPS/(Frame/s) |
|---|---|---|---|
| PSPNet (Mob) | 2.12 | 2.38 | 141.16 |
| PSPNet | 41.60 | 46.71 | 57.09 |
| U-Net (vgg) | 176.43 | 24.89 | 25.46 |
| U-Net (resnet50) | 71.31 | 43.93 | 70.97 |
| HRNetv2_w32 | 32.00 | 29.54 | 22.77 |
| Deeplabv3plus | 58.33 | 54.71 | 75.29 |
| HRNetv2_w48 | 66.17 | 65.85 | 20.10 |
| HRNetv2_w18 | 13.06 | 9.64 | 24.91 |
| Ours | 2.28 | 2.23 | 117.68 |

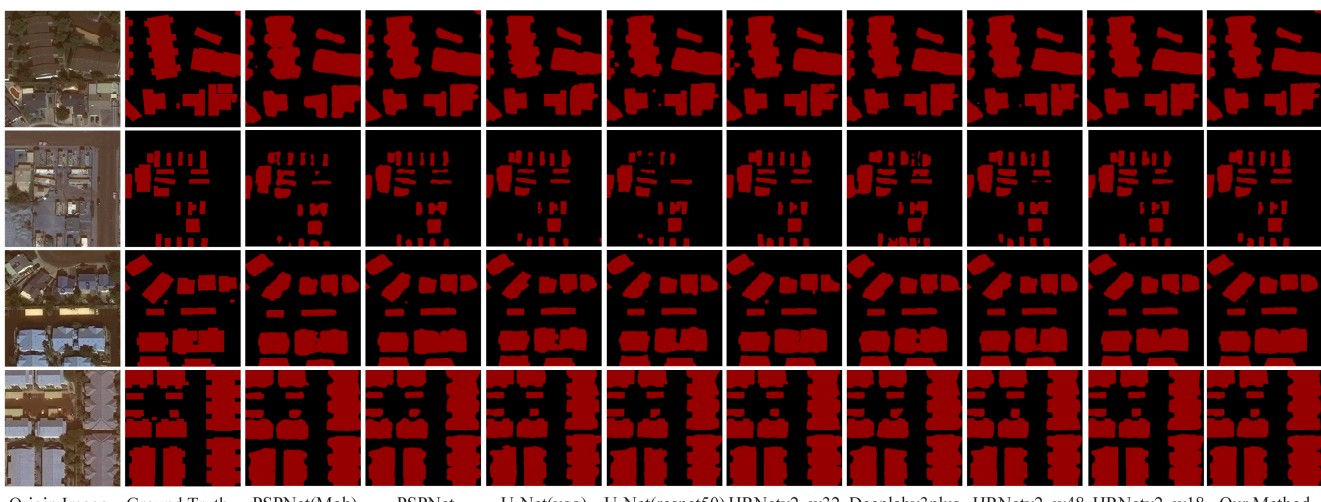

Origin Image　Ground Truth　PSPNet(Mob)　PSPNet　U-Net(vgg)　U-Net(resnet50)　HRNetv2_w32　Deeplabv3plus　HRNetv2_w48　HRNetv2_w18　Our Method

**Figure 7.** Comparative experimental results for the Mapchallenge building dataset.

On the dataset featuring typical urban architectural instances in China, according to Tables 5 and 6, the proposed improved model achieves the highest accuracy and faster prediction speed in urban building extraction compared to other algorithms. Specifically, the mIoU reaches 85.11%, the PA is 86.35%, the mPA is 91.61%, and the FPS rate is 110.67. As shown in Figure 8, the improved algorithm demonstrates superior delineation of edge contours for buildings of varying shapes compared to other algorithms. Furthermore, it addresses the issue of misidentification of buildings. However, contrasting algorithms still exhibit instances where the contours of buildings intersect with those of non-buildings, whereas the improved method effectively highlights diverse building edge contours and minimizes interference from non-building objects to a greater extent.

**Table 5.** Comparative experimental results (from the Typical Chinese Urban Buildings dataset).

| Models | IoU (Building)/(%) | mIoU/(%) | PA/(%) | mPA/(%) |
|---|---|---|---|---|
| PSPNet (Mob) | 65.37 | 77.41 | 78.08 | 86.53 |
| PSPNet | 73.65 | 83.01 | 84.75 | 90.47 |
| U-Net (vgg) | 75.12 | 84.17 | 87.47 | 91.75 |
| U-Net (resnet50) | 75.38 | 84.51 | 83.86 | 90.57 |
| HRNetv2_w32 | 74.46 | 83.88 | 85.07 | 90.83 |
| Deeplabv3plus | 75.61 | 84.47 | 88.11 | 91.63 |
| HRNetv2_w48 | 76.02 | 84.86 | 85.63 | 91.29 |
| HRNetv2_w18 | 76.09 | 84.80 | 88.05 | 92.13 |
| Ours | 76.44 | 85.11 | 86.35 | 91.61 |

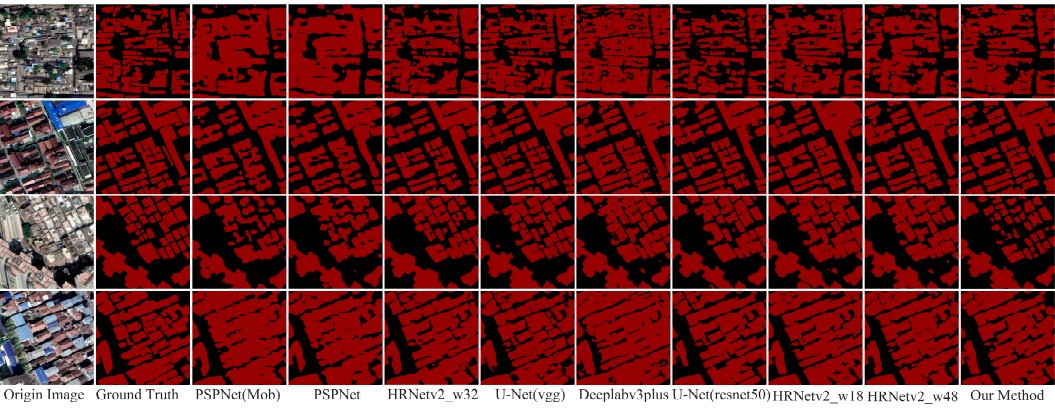

Origin Image　Ground Truth　PSPNet(Mob)　PSPNet　HRNetv2_w32　U-Net(vgg)　Deeplabv3plus　U-Net(resnet50)　HRNetv2_w18　HRNetv2_w48　Our Method

**Figure 8.** Comparative experimental results of the Typical Urban Buildings in China dataset.

**Table 6.** Comparison of the individual model parameters in the Typical Chinese Urban Buildings dataset.

| Models | Gfloaps/(GB) | Params/(MB) | FPS/(Frame/s) |
|---|---|---|---|
| PSPNet (Mob) | 6.03 | 2.38 | 154.32 |
| PSPNet | 118.43 | 46.71 | 57.09 |
| U-Net (vgg) | 451.67 | 24.89 | 24.47 |
| U-Net (resnet50) | 184.10 | 43.93 | 35.25 |
| HRNetv2_w32 | 88.34 | 29.54 | 22.77 |
| Deeplabv3plus | 166.84 | 54.71 | 40.64 |
| HRNetv2_w48 | 182.61 | 65.85 | 21.01 |
| HRNetv2_w18 | 36.11 | 9.64 | 20.63 |
| Ours | 6.51 | 2.23 | 110.67 |

## 4. Discussion

According to the ablation experiments, the improved model not only improves the accuracy of building extraction, but also significantly improves the computational efficiency of the model. In comparison with the baseline model, the improved model increases the mIoU by 0.64%. As indicated in Tables 1 and 2, employing Mobilenetv2 as the backbone network significantly boosts the model's prediction speed, increasing FPS by 96.89, but results in a 0.34% decrease in mIoU. This suggests that while Mobilenetv2 improves the model's prediction speed, it slightly compromises accuracy in feature extraction. In addition, as can be seen in Figure 6, the improved algorithm incorrectly segmented non-building entities and omitted small buildings when using Mobilenetv2 as the backbone network, suggesting that the algorithm's ability to extract the features of small buildings is insufficient, meaning, in turn, that it fails to extract small building contours. The introduction of the coordinate attention (CA) mechanism enables the network to learn the coordinate information of different locations in the input image, focusing on the exact location of the building in space, which in turn better captures the edges and contour features of the building, resulting in an increase of 0.27% in mIoU. Figure 6 shows that although the occlusion problem (low-rise buildings are occluded by high-rise buildings and buildings are occluded by trees) leads to missing building contours, the CA mechanism recovers the building contours in the regions occluded by high-rise buildings and trees by enhancing the attention to the building's own features (e.g., edges, textures, which are related to the building contours). However, in the process of enhancing attention to the building features, the CA mechanism pays too much attention to some local features and thus misextracts non-building features, which indicates that the mechanism is more efficient in forming building contours but slightly less effective in overcoming mis-extractions of non-building features. As shown in Figure 6 and Table 1, the introduction of the SP module in ASPP enhances the model's ability to extract building edges by capturing features at building edges more efficiently, resulting in a more accurate building profile. Meanwhile, the SP module adopts an appropriate pooling scale to limit the feature extraction range of the improved model under a larger receptive field, reducing the focus on non-buildings, which in turn significantly reduces the mis-extraction of non-building feature elements, resulting in an improvement of 0.52% in mIoU. Compared with the previous two improvement strategies, the SP-ASPP module shows the greatest increase in accuracy, indicating that the SP-ASPP structure achieves a more complete building outline extraction. The above improvement strategies indicate that MobileNetV2 significantly enhances model computational efficiency and prediction speed, while the other two modules effectively segment urban buildings. According to Table 1 and Figure 6, the improved method achieves the highest extraction accuracy, and there are no obvious false positives or problems such as discontinuous building contours, which proves the effectiveness and accuracy of the improved strategy.

In comparative experiments, PSPNet, U-Net, Deeplabv3plus, and HRNetv2 were employed as mainstream algorithms for evaluation. According to Tables 3 and 4, on the Mapchallenge dataset, the improved model achieves the highest precision in building extraction and the fastest prediction speed among the compared algorithms. As depicted in Figure 7, contours of buildings extracted by the comparative algorithms appear discontinuous, whereas the improved algorithm produces smoother building contours without obvious jagged edges, demonstrating its capability to effectively address the issue of discontinuity in building contour edges. Additionally, the PSPNet algorithm exhibits false positives of non-building entities, indicating that the improved algorithm can achieve precise building extraction without introducing issues such as false positives. Furthermore, based on Figure 7, HRNetv2_w32 and the Mobilenet-PSPNet algorithm exhibit irregularities in the contours of small-sized buildings, incompleteness in the contours of medium-sized buildings, and unclear contours of large-sized buildings when extracting multiscale buildings. In contrast, the improved algorithm can clearly extract the contours of buildings at different scales, which affirms the effectiveness of the proposed method. According to Tables 5 and 6, on a typical dataset of urban buildings in China, the improved algorithm achieves an mIoU of 85.11%, and in terms of model prediction speed, it achieves 110.67 FPS. This confirms the effectiveness and efficiency of the improved model, making it suitable for practical deployment and meeting real-world demands. Figure 8 illustrates that the comparative algorithms encounter issues such as missing extractions for small-sized buildings and contour intersections for densely arranged buildings. In contrast, the improved algorithm can clearly extract the edge contours of various types of buildings without causing edge overlaps, showcasing its ability to achieve fine-grained extraction of building contours. Simultaneously, within a more straightforward urban context, the improved algorithm's building extraction results tend to be more realistic compared to those of PSPNet and HRNet_w32. It minimizes interference from non-building entities, achieves more accurate contour extraction for irregular buildings, and demonstrates the stability of the improved algorithm in distinguishing between buildings and non-buildings. In summary, the improved algorithm effectively accomplishes the building extraction task and, compared to other mainstream semantic segmentation algorithms, exhibits higher speed and accuracy.

Furthermore, although the improved algorithm demonstrates a significant increase in prediction speed, the improvement in extraction accuracy is less than 1%. This is attributed to the excessive simplification of the model structure after lightweighting, reducing the model's complexity and consequently impacting its precision. Additionally, it is noteworthy that the improved algorithm solely addresses the task of urban building extraction from remote sensing images, lacking experimental validation on other computer vision tasks. Therefore, in future research endeavors, from a model optimization perspective, we can contemplate algorithm refinement by considering the complexity of the model structure. From an application standpoint, exploring the adaptability of this algorithm to diverse computer vision tasks could be pursued through techniques such as transfer learning.

## 5. Conclusions

Aiming to confront the challenge that it is difficult to completely extract the outlines of urban buildings from high-resolution remote sensing images due to occlusion, mixing and other problems, a more efficient and accurate deep learning network is proposed to achieve building extraction. The coordinate attention module is used to enhance the attention to buildings' contours in different positions to extract more complete building contours. The SP-ASPP module is used to further obtain the building edge feature information and improve the building extraction accuracy. Experimental results on datasets comprising typical architectural developments in Chinese cities and on the Mapchallenge building dataset demonstrate that the improved model achieves higher extraction accuracy. Ablation experiments reveal that the three improvement strategies of the enhanced model effectively enhance capacity for urban building extraction. Additionally, the introduction of the SP-

ASPP module increases the model's mIoU from 84.47% to 84.99%, effectively enhancing urban building extraction accuracy. Furthermore, with the integration of the lightweight Mobilenetv2 backbone, the model's prediction speed escalates from 40.64 FPS to 137.53 FPS, enabling real-time response and practical deployment in engineering projects. Notably, combining Mobilenetv2 with the SP-ASPP and CA modules further improves the urban building extraction accuracy to 85.11%, while achieving a speed of 110.67 FPS. In the future, the combination of spectral features in remote sensing images (e.g., vegetation index, water index, building index, etc.) with deep learning models can also be explored to further improve the extraction efficiency of urban buildings.

**Author Contributions:** J.H., Z.R. and Y.C. designed the study. Y.C. and J.H. conducted the analysis and wrote the manuscript. W.W., W.Z., C.Z. and J.H. put forward improvement advise. Y.C., J.H. and Z.R. interpreted the results and revised the manuscript. All authors have read and agreed to the published version of the manuscript.

**Funding:** This research is supported by the National Natural Science Foundation of China under grant no. 41975183, the Innovation Project of LREIS (YPI007), and grant no. 41875184.

**Data Availability Statement:** The dataset utilized to substantiate the findings of this study can be obtained from the authors upon request. Detailed information regarding the dataset, including its composition and characteristics, can be found in the references provided, specifically reference [39].

**Acknowledgments:** We thank the anonymous reviewers for their comments and suggestions that improved this paper.

**Conflicts of Interest:** The authors declare no conflicts of interest.

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
