# Peer review of "A Lightweight Building Extraction Approach for Contour Recovery in Complex Urban Environments"

_remotesensing, doi:10.3390/rs16050740_

Round 1

Reviewer 1 Report

Comments and Suggestions for Authors

This manuscript addresses building extraction from remotely sensed imagery with a lightweight network, and uses the coordinate attention module, and pooling fusion module.However, I believe that MobileNetv2, ASPP, and Coordinate Attention (CA), which are used in the manuscript, have all been widely used. and proven in the field of remote sensing imagery. This manuscript does not contribute anything new.

In addition, the followingcomments would like to be considered:

1.Figures 1 to 3 suggest that each layer is labeled with its size (C*H*W).

2. the results section focuses more on comparison experiments with other networks, can we add effect fusion experiments to prove the contribution and impact of each module in our proposed method.

3. If the highlight is to highlight the lightweight, in addition to FPS, the number of parameters should be analyzed to highlight the advantages of the network.

4. equations 1-3 lack the description of each variable.

5. the text portion in Figures 6 through 8 is not clear enough.

Reviewer 2 Report

Comments and Suggestions for Authors

The work is interesting and it was pleasure to read.
Did there be a performance difference between Orthorectified and non-orthorectified images to ensure the diversity of the data set? On the other hand, is the performance success of the images with dense buildings and less dense buildings?

Table II and Table III The layout of the names of models can be the same as each other.

Finally, I encourage the authors to make the dataset (and the training-test splits) publicly available. 

Reviewer 3 Report

Comments and Suggestions for Authors

In the study, a new deep learning network is proposed for building extraction. The manuscript is generally well organized and has interesting results. Here are some revision suggestions:

1) It would be more appropriate to include suggestions for future studies in the Abstract and in the Conclusion.

2) Current studies can be added to expand the literature research and especially for the DeepLab approach. Some suggestions:

Lin, H., Hao, M., Luo, W., Yu, H., & Zheng, N. (2023). BEARNet: A novel buildings edge-aware refined network for building extraction from high-resolution remote sensing images. IEEE Geoscience and Remote Sensing Letters.

Atik, S. O., Atik, M. E., & Ipbuker, C. (2022). Comparative research on different backbone architectures of DeepLabV3+ for building segmentation. Journal of Applied Remote Sensing, 16(2), 024510-024510.

3) Do the study results include only the results of the building class? General metrics without distinction between building and non-building? In two-class classifications, non-building generally has a higher metric, increasing overall accuracy. Seeing class-based results will be useful to evaluate building extraction performance.

4) What is the advantage of the proposed method being faster? For example, if it is the number of parameters, it is expected to be presented in a table.

Reviewer 4 Report

Comments and Suggestions for Authors

Dear authors,

Thank you for your efforts in producing this paper. In general, the paper is well written, my comments are listed below:

Figure 1 – some of the texts are hard to read, the authors may consider using bigger fonts.

Comment for the methods part: This part is relatively well described, but I would recommend expanding it a bit and describing the individual steps in more detail.

Table 1: I recommend slightly modifying Table 1, it may be difficult for the readers to understand, e.g. to which data the individual lines belong.

General comment and the main issue, that should be discussed further: Based on the results, and also the authors state in lines 410-411: “…the improvement in extraction accuracy is less than 1%.”, so the benefit of the author’s described method is rather an increase in efficiency?? Since, as the authors state also, there is not a big difference in the accuracy. So, it is not entirely clear to me what the main contribution of the article is.

Regards

Comments on the Quality of English Language

Minor editing of English language required.

Round 2

Reviewer 1 Report

Comments and Suggestions for Authors

This manuscript has been improved over the previous version and answers my questions. In particular, the quality of the images, and the descriptions of the formulas have been improved across the board, which will not give the reader the wrong idea that this work is complete in its own right at this time. And my question now is whether MobileNetv2, ASPP, and Coordinate 

Attention (CA) are still well established methods, they were proposed a few years ago, refer to the following literature:

(1) https://arxiv.org/abs/2103.02907

(2) https://arxiv.org/abs/1606.00915

Even in the field of building extraction, these strategies have been validated, cf:

A coarse-to-fine boundary refinement network for building footprint extraction from remote sensing imagery,

ISPRS Journal of Photogrammetry and Remote Sensing, https://doi.org/10.1016/j.isprsjprs.2021.11.005.

There is also insufficient discussion on why these methods are adaptable to the difficult issues raised by the authors such as the occlusion problem for building extraction. It is recommended that the authors and editors further consider whether this meets the requirements of the journal, and if it is felt that this work meets the goals of the journal, the process can be taken to the next step.
